# Recovery of Naringin-Rich Flavonoid Extracts from Agroresidues with Anxiolytic- and Antidepressant-like Effects in Mice

**DOI:** 10.3390/molecules27238507

**Published:** 2022-12-03

**Authors:** Liliana Hernández-Vázquez, Julia Cassani, Ivo Heyerdahl-Viau, Rubria M. Martínez-Casares, Héctor Luna, Ana María Dorantes-Barrón, Daniel Arrieta-Báez, Rosa Estrada-Reyes

**Affiliations:** 1Departamento de Sistemas Biológicos, Universidad Autónoma Metropolitana, Unidad Xochimilco, Ciudad de México C.P. 04690, Mexico; 2Laboratorio de Fitofarmacología, Dirección de Investigaciones en Neurociencias, Instituto Nacional de Psiquiatría Ramón de la Fuente Muñiz, Calzada México-Xochimilco 101, Col. San Lorenzo Huipulco, Tlalpan, Ciudad de México C.P. 14370, Mexico; 3Instituto Politécnico Nacional, Centro de Nanociencias y Micro y Nano Tecnologías, Unidad Profesional Adolfo López Mateos, Ciudad de México C.P. 07738, Mexico

**Keywords:** naringin, flavonoids, anxiety, depression, organic waste, agroresidues

## Abstract

*Citrus paradisi* species belong to the Rutaceae family, and it is commonly known as grapefruit. Grapefruit consumption involves a large amount of waste that goes to landfills and produces significant pollution affecting the human health. To examine this phenomenon, we designed an efficient chemical method that recovers naringin-rich flavonoid extracts from the fresh waste of grapefruits, by using the solvent impregnation resin method (SIR) with XAD-4 amberlite and either methanol or water as elution systems. Additionally, we focused on evaluating these extracts’ anxiolytic- and antidepressant-like effects in behavioral predictive paradigms in mice. According to direct Principal Component Analysis (PCA) by NMR, and Direct Injection Electrospray Ionization-Mass Spectrometry (DIESI-MS), methanol extracts obtained after resin treatment were free of coumarin compounds and evinced had a high content of naringin. Poncirin, phenylalanine, chrysin 5,7-dimethyl ether, 5,7-dimethoxy-4′-hydroxyflavanone, 2,3-dihydro-2-(4-hydroxyphenyl)-5,6,7,8-tetramethoxy-4H-1-benzopyran-4-one, tetrahydrocurcumin, corchoionoside C, 6′-coumaroyl-1′-*O*-[2-(3,4-dihydroxyphenyl) ethyl]-*β*-D-glucopyranoside were also detected. Naringin-rich methanol extract caused a clear anxiolytic-like effect in the Elevated Plus Maze (EPM) and the Hole-Board (HBT) Tests, increasing oral doses of this extract did not produce a sedative effect. A single oral dose caused an antidepressant-like effect in the Tail Suspension Test (TST), while repeated administrations of the methanol extract elicited a robust antidepressant effect in the Forced Swimming Test (FST) in mice. Our evidence highlights the importance of bioprospecting studies of organic waste with therapeutic potentials, such as anxiety and depression disorders.

## 1. Introduction

*Citrus paradisi* belongs to the Rutaceae family, commonly known as grapefruit is considered a functional food; its nutrient content represents is a valuable source of vitamin C, carbohydrates, dietary fiber, various B vitamins, minerals, as well as pharmacologically active compounds, such as carotenoids, and free (aglycon) and glycosylated flavonoids [1].

Grapefruit is a natural hybrid between pomelo and sweet orange. Its first mention as a species was in 1750, on the Caribbean Island Barbados. From this date onwards, grapefruit has been diversified by spontaneous mutations so that the botanic identity of commercialized species is complex. However, they have preserved high heterozygosity, exceptionally low genetic variability, and the metabolism profile of the *Citrus* species has been conserved. Among the grapefruit’s secondary metabolites, polyphenolic and flavonoid compounds have shown a wide variety of pharmacological activities, such as antioxidant and anti-inflammatories [2], cardioprotective [3] effects anticancer [4], antiviral [5], as neuroprotective agents [6] among other [7] and, particularly, their anxiolytic- as well as antidepressant-like effects have proved to be outstanding [8,9,10].

In this regard, it is important to note that anxiety and depression are the most prevalent neuropsychiatric diseases worldwide, leading causes of global health burden (Lancet 2021). Because of this, the search for safe and accessible alternatives to alleviate these disorders is crucial.

Grapefruit species are one of the citric fruits most consumed globally; around four million metric tons are produced annually. However, after consumption, its residues end up in landfills, causing a large amount of agroresidues and negatively impacting the environment [11]. Garbage, such as peels and albedo (the bitter white part of the *Citrus* peel), are rich in flavonoid-type compounds. It is noteworthy that naringin (flavanone-7-*O* glycoside) is one of their main constituents. We paid particular attention to naringin due to its unique pharmacological activities such as antianxiety and tranquilizer agent. Substantial evidence supports their use in treating mood disorders and highlights other beneficial health actions [12]. In this regard, agro residual of grapefruit could be a new natural source of available and inexpensive biomolecule obtention with therapeutic potential for mood disorders. Different methods have been tested to isolate flavonoids using Solvent-Impregnated Resins (SIR) techniques such as XAD-2, XAD-4, and XAD-16 [13,14,15]. Amberlite XAD-4 is a polymeric resin or more exactly a microporous polystyrene-divinylbenzene copolymer, which acts as a good polymeric absorbent and is used for stripping and concentrating organic compounds. Usually employed under isocratic conditions, Amberlite XAD-4 serves to remove chlorinated organics, pesticide traces, metals, or any contaminant traces [15]. However, to our knowledge, whether these methods are helpful for recovering naringin and flavonoid compounds from fresh waste generated by grapefruit consumption has not been explored.

In this study, we evaluated the amberlite resin (XAD-4) impregnation method to recover flavonoid-rich extracts, especially naringin, from grapefruit waste and remove sugars or any inorganic waste contaminant. For the flavonoid metabolic profile of extracts, we used the DIESI-MS detection and multivariate statistical analyses, such as Principal Component Analysis (PCA) and Orthogonal Projections to Latent Structures Discriminant Analysis (OPLS-DA) by Nuclear Magnetic Resonance. Additionally, we focused on evaluating naringin-rich extract anxiolytic- and antidepressant-like effects in behavioral predictive paradigms in mice. We analyzed the anxiolytic-like effect in the Elevated Plus-Maze Test. Sedative effects were assessed on Hole-Board Test. Antidepressant-like effects of the naringin-rich extract were evaluated in the Tail Suspension and the Forced Swimming Tests.

## 2. Results

### 2.1. Naringin and Flavonoids Rich Extracts

To obtain extracts with high naringin and flavonoid content, two protocols were conducted after amberlite SIR. Briefly, independent samples of the crude extracts (three samples with three replicates each one; E1, E2, E3) were impregnated with XAD4 resin in methanol and, in a first approach, the impregnated samples were eluted with methanol (M1) followed by a second elution with water (W1). The extracts were dry, and the percentage yield was calculated as the dry weight against the dry weight of grapefruit waste. Our results are shown in Table 1.

The average yield was 10.6 g corresponding to a yield of 35.33% concerning to the dry weight of grapefruit waste for methanol fractions, and 0.76 g corresponding to 2.5% in relation to the dry weight for water fractions.

In a second approach (protocol 2), the impregnated samples were recovered (three samples with three replicates, each one; E1, E2, E3) with water first (W2) and afterwards eluted with methanol (M2). Results and recovery percentages are shown in Table 2.

The average yield was 10.39 g, representing 34.8% in relation to the dry weight and for water extraction, and 1.34 g corresponding to a yield of 4.53% in relation to dry weight for methanol extract.

Comparing protocols of recovering naringin-rich extracts, as shown in Table 1 and Table 2, entries M1 and W2, most the constituents’ desorption from Amberlite XAD-4 resin eluted in the first fraction. However, naringin and flavonoids are recovered in methanolic extracts M1 and M2, as we analyze them through MS and NMR spectra.

From these results, Protocol 1, which had good reproducibility and a high recovery percentage, was the best method for obtaining naringin and flavonoids rich extract from dry peel from grapefruit waste.

### 2.2. NMR Proton Spectra

^1^H NMR experiments were conducted to compare the different components present in methanolic fractions before and after recovering flavonoids from amberlite.

Figure 1 shows three main regions: aromatics, glycoside, and flavanones epimeric regions. Signals from 6 to 8.5 ppm correspond to aromatic protons systems characteristic of flavonoids. The intensity of these signals significantly increased after SIR treatment for methanol extracts; whereas intensity of this signal decreased in water extracts due to the increased signal in the sugar regions from 3.8 to 4.3 ppm. Relation to the anomeric region, around 5 ppm, these signals increased in water fractions and decreased in methanol fractions for both protocols. The epimeric region around 5.5 ppm attributable to the proton at C-2 which is characteristic of flavanones such as naringin and poncirin these are visible after SIR treatment for the methanolic fraction. It is noteworthy that the signals corresponding to anomeric sugars regions showed a low intensity in methanolic fractions M1 and M2, compared to the methanolic extract without SIR treatment.

### 2.3. Principal Component Analysis (PCA)

In order to show a differentiation between the two groups, a multivariate statistical method was performed, in which aqueous and methanolic fractions were compared after they were recovered from Amberlite XAD-4 resin. Even though the NMR data in Figure 1 showed a clear difference between the two groups, we performed a PCA to further asses this difference and analyze the variance.

We analyzed twenty samples from different recollection sites were selecting randomly from among all the methanolic and water fractions of both protocols. Ten samples of the water and ten of the methanolic fractions were taken from the SIR procedure. PCA was applied to whole signals from the ^1^H NMR spectra, showing clear discrimination between both fractions. Data variances were explained with the cumulative percentage of the two first components (PC1 75.6% and PC2 13.7%, Appendix A). Those results showed selectivity in the resin impregnation to flavonoids, for instance naringin, over other components. As shown in Figure 2, the OPLS-DA showed a clear separation of samples of water and methanol fractions. These results were confirmed by ^1^H NMR spectral, which showed that both groups have a clear differentiation due to their whole NMR signals. Furthermore, the protocol is highly reproducible independently of the Grapefruit sources, it had values of Q^2^ 0.873, which makes it a good predictive model for samples not included in it and a value of R^2^Y of 0.931, which allows for a good model adjustment.

### 2.4. DIESI-MS Analysis

Analysis of contained flavonoids in M and W fractions was conducted by DIESI-MS; as shown in Table 3, for the methanol extract (without SIR-treatment, EM), the percent of Relative Area (%RA) of naringin was 0.8, while for M extract treated with SIR methodology in protocol 1, the naringin %RA was 10.5. Similar increases were observed in poncirin, which is a flavanone rutinoside. Without treatment extract had a %RA of 0.3% and after treatment with XDA-4 an increase to %RA 3.0. These results agree with our findings from the NMR analysis. Together, our results showed the metabolic profile of the extract of grapefruit waste after SIR treatment. The main constituents recovered from SIR method are shown in Figure 3.

### 2.5. Behavioral Evaluation. Pharmacological Evaluation

Anxiolytic-like effect of a single oral dose of naringin-rich extracts was evaluated in the elevated plus-maze test at 25, 50, 100, 200, and 400 mg/kg doses. Our results showed that the extract significantly increased the exploratory behaviors in the elevated plus-maze test.

Figure 4 show the naringin-rich extract obtained by Protocol 1.

Results indicated that M, with 34.7% recovery flavonoids content (10.40 g) increased the time spent in the open arms (H = 34.2, fd = 5, *p* ≤ 0.001), reaching the best effect at 50 and 100 mg/kg; at 200 mg/kg this effect diminished concerning to the control group, and the highest dose (400 mg/kg) did not produce changes in the time spent in the open arms. The extract produced a significant decrease in the time spent in the closed arms in comparison to the control group (H = 36.84, fd = 5, *p* ≤ 0.001).

In a similar form, Diazepam (DZ) increased the time spent on the open arms with a simultaneous reduction in the time spent in the closed arms (H = 32.292, fd = 4, *p* ≤ 0.001; H = 31.410, fd = 4, *p* ≤ 0.001, respectively).

Evidence shows that flavonoid compounds can produce sedation when administered at high doses, but this information is controversial [11]. We decided to evaluate the sedative effect of single oral administration with increasing doses of the naringin extract in the Hole Board Test. Our results showed that the extract at 200, 400, 600, 800, and 1000 mg/kg did not produce changes in the counts’ number (H = 15.78, fd = 5, *p* = 0.007); in contrast, DZ at 4 mg/kg significantly decreased the count number to compared to the control group (t = 74.0, *p* = 0.004), as opposed to the extract that, in comparison to the control group increased both the head dipping time (H = 28.64, fd = 5, *p* ≤ 0.001) and the number of rearing (H = 25.24, fd = 5, *p* ≤ 0.001) (Figure 5). In summary, the methanol extract of Grapefruit did not modify the ambulatory activity or increase the exploratory behavior, and it did not produce sedative effects in the experimental mice. Furthermore, the hole board test was sensible for detecting the anxiolytic actions of the methanol extract.

The antidepressant-like effect of the naringin extract (M1) was evaluated in the tail suspension test (TST). As shown in Figure 6A, a single oral dose of extract at 100 and 200 mg/kg significantly reduced the immobility time compared to the control group and in a similar manner to imipramine (IMI) at 25 mg/kg (F_(5, 47)_ = 105.21, *p* ≤ 0.001).

In the FST, it was necessary to administer reaped doses (a once-daily dose for seven days; 25, 50, 100, and 200 mg/kg) of the extract (Figure 6B), to significantly reduce the immobility time (F_(5, 46)_ = 31.81, *p* ≤ 0.001) in comparison to the control group and in a similar manner to fluoxetine (FLX) at 5 mg/kg

The ambulatory activity of animals subjected to the TST and the FST was evaluated in the Open Field Test (OFT) to discard possible negative or positive false effects.

Our results showed that none of the naringin extract M1, IMI or FLX treatments affected the ambulatory activity of experimental subjects. Thus, the reduction in the immobility behavior can be attributed to an antidepressant-like effect produced by the grapefruit extract waste (Appendix A).

## 3. Discussion

Nowadays, there is a growing environmental awareness regarding waste generation; agro-residues are generated by tons during processes of food consumption [16]. The production of added-value compounds from organic waste has become an essential line of research. Agro-waste obtained from fruit extracts, plant fiber, and fruit peels is a great green resource for producing valuable products through efficient recovery protocols [17]. Besides, the use of agro-residues could minimize pollution produced by them and reduce the negative environmental impact.

*Citrus* species contain a great diversity of compounds such as vitamins and minerals; therefore, they can be considered functional food. Evidence indicates that *Citrus* waste contains phenolic, and free (aglycon) and glycosylated flavonoids. Especially, the flavanone glycoside, naringin, is the major component of *Citrus* species and a chemotaxonomic marker of the *Citrus* genus. Furthermore, it is a prototype compound for studying new drugs due to its extraordinary pharmacological actions and health benefits, such as antinociceptive, anti-inflammatory, neuroprotective, and modulator memory impairment and anxiety [18,19].

Given these properties, agro-residues produced by grapefruit waste represent a useful and inexpensive source of active principles with therapeutic potential for mood disorders.

In this study, we performed a bio-recovery approach to produce efficient and inexpensive naringin-rich extracts from fresh grapefruit peel residues with potential application in treating anxiety and depression disorders.

Two approaches were employed to recover the flavonoids, especially naringin, from fresh grapefruit waste. Our findings showed that the SIR method using methanol as a desorption and elution agent resulted in flavonoid-rich extracts (yield of 35% in relation to concerning dry grapefruit waste). The OPLS-DA clearly showed that samples can be separated in two groups, to avoid an overfitting of the model we used five permutations for getting a good adjustment of the model with a difference between R^2^Y and Q^2^ > 0.05. These results indicate that methanol and water extracts have significantly distinct metabolic profiles. ^1^HNMR experiments allowed us to analyze the different components present in water, as well as methanol fractions before and after the SIR treatment. Our results showed that naringin-rich methanol significantly increases the signals of the aromatic protons, particularly those corresponding to an AA’BB’ system characteristic of ring C of flavonoid compounds (naringin and poncirin), and epimeric region, around 5.5 ppm for naringin as a double of doubles corresponding to a proton attached to C-2 and the equivalent proton attach to C-2 for poncirin at 5.6 ppm. Furthermore, the intensity of anomeric signals in the sugar region in methanol extracts decreased. These results indicate that the methanol elution was efficient in the glycosides extraction. PCA applied to NMR with unsupervised statistical analysis allowed us to identify differences between the water and methanol fractions SIR obtained from the two extraction protocols. NMR mixture analysis allowed us to determine the identity of different compounds without purifying them. Furthermore, this analysis showed that flavonoid metabolic profiles were similar regardless of the origin of the grapefruit waste.

Methanol extract without SIR treatment, water and methanolic fractions were subjected to an analysis DIESI-MS to determine their metabolic profile using direct-injection electrospray ionization mass spectrometry. Thus, DIESI-MS analysis revealed the presence of naringin in 10% of the relative area (%RA), namely, tenfold more of its content in extracts without resin treatment (0.8%RA of naringin). This analysis also showed the presence of phenylalanine, chrysin 5,7-dimethyl ether, 5,7-dimethoxy-4′-hydroxyflavanone, 2,3-dihydro-2-(4-hydroxyphenyl)-5,6,7,8-tetramethoxy-4H-1-benzopyran-4-one, tetrahydrocurcumin, corchoionoside C, 6′-coumaroyl-1′-O-[2-(3,4-dihydroxyphenyl) ethyl]-*β*-D-glucopyranoside, naringin and (2*S*)-poncirin. Hence, this was a robust analytical method for enhanced flavonoids content from Grapefruit residues.

On other hand, in the last decades, evidence has shown the therapeutic potential of flavonoids compounds by CNS actions and strongly suggests that these natural products are interesting prototype compounds for the study new anxiolytic and antidepressant drugs [18,19,20].

We focused on evaluating naringin-rich extract anxiolytic- and antidepressant-like effects in behavioral predictive paradigms in mice.

Recently, the central actions of naringin in behavioral models in mice were explored [21,22] and, reported the antianxiety-like effect in mice that were treated with increasing intraperitoneal doses of naringin (2 to 10 mg/kg). In this work, we found an anxiolytic-like effect with an oral route administration of the naringin extract (25 to 400 mg/kg). The elevated plus maze offers a simple method for measuring the response of mice to an unfamiliar environment, with the advantage that behaviors can be observed and quantified. For example, the principal behavior that gives information about anxiety levels is the time spent exploring the maze’s open arms; the increase in this exploratory behavior indicates low anxiety levels. It is well known that anxiolytic drugs such as Diazepam (DZ), a benzodiazepine (BDZ) of clinical use, increase the exploratory behaviors in the EPM test, but high doses of it, produce unseeable side effects, such as sedation and myorelaxation [23]. Hence, our results showed that the naringin-rich extract significantly increased the exploratory behaviors and reached the best effect at 200 mg/kg.

Furthermore, in contrast to DZ, animals treated with the naringin-rich extract at high doses (200 and 400 mg/kg) did not show behaviors that denote sedation, loss of muscle tone, or impairment in the ambulatory activity. The EPM test is also sensitive to changes in the ethological behavior of the animals such as head dipping and stretched behavior denoting uninhibited (anxiolysis) behavior in face of the challenge imposed by the maze’s height. The extract causes a significant increase in this ethological behavior in the EPM test. In addition to its anxiolytic effects, evidence indicates that flavone and flavanone glycosides can induce sedative effects. In this regard, it has been reported that naringin high doses administered by intraperitoneal route, besides their anxiolytic effect (1 y 3 mg/kg), also produce a significant sedative effect at 30 mg/kg [17].

Based on this, we evaluated the possible sedative effects of the naringin-rich extract in the Hole Board test, which can detect both anxiolytic and sedative effects. Our results showed that the naringin-extract at doses above 200 mg/kg until 1000 mg/kg increased the exploratory behaviors without modifying the ambulatory activity. Thus, neither of the doses evaluated did cause the sedative effects or behavioral changes. In contrast, diazepam at 4 mg/kg significantly decreased exploratory behaviors and ambulatory activity, caused thus a sedative effect in the hole board test. It is worth mentioning that, in contrast to what [17,24] reported, the naringin-rich extract in different dosages neither induced a sedative effect nor affected the ambulatory activity of mice. In addition, the oral administration route could represent an advantage in the treatment accessibility.

We also evaluated the antidepressant-like effects of the naringin extract in two predictive behavioral models. Our results showed that a single oral administration with 200 and 400 mg/kg of the naringin-rich extract produced a clear antidepressant-like effect in the tail suspension test. In contrast, in the forced swimming test, administration with single doses of the extract was not enough to produce the antidepressant-like effect (data not shown). Thus, we decided to assess the repeated oral administration in the forced swimming test, similar to that reported by Ben-Azu and coworkers (2019) [21]. We found that a single oral dose once a day for seven continuous days caused a robust antidepressant-like effect in the forced swimming test.

It is known that amphetamines can decrease immobility without having an antidepressant-like effect, or sedative drugs can increase this passive behavior [25]. Our results, however, showed that none of the treatments affected the mice’s locomotor activity. Thus, the immobility behavior reduction can be attributable to antidepressant-like specific actions of experimental drugs.

Our result showed that, as expected, a single oral treatment with the naringin-rich extract produced a robust anxiolytic-like effect in mice. Furthermore, the naringin extract elicited an antidepressant-like effect in the tail suspension test, while repeated doses of the naringin-rich extract caused a robust antidepressant-like effect in the forced swimming test. Without a doubt, we can attribute these effects to naringin. However, we previously reported about the anxiolytic actions of neoponcirin in mice, an isomer of poncirin [6], and in this report we did not discard the participation of the poncirin, and other flavonoids present in the methanol extract of Grapefruit waste in the anxiolytic actions of the extract of grapefruit waste. Altogether our results showed that agro-residues from grapefruits could be a valuable source of principles active with anxiolytic and antidepressant properties.

## 4. Materials and Methods

### 4.1. Grapefruit Wastes Recollection and Extracts Obtention

Grapefruit (*Citrus paradisi*) wastes were collected from different local markets in Mexico City from August 2019 to November 2021. The wastes were washed with water, placed on an absorbent towel to remove excess water, and kept in refrigeration. Afterwards, the peels (flavedo and albedo) were recovered and finely cut into small pieces, frozen, and dried in a Telstar Cryodos (model CRYODOS-50) freeze dryer at −50 °C and 0.01 mBar for 8 to 10 h. Dried peels were finely ground in a blender and separated into homogeneous samples of 30 g each. Samples (30 g) were macerated in 300 mL of methanol (HPLC grade; in 1:10 weight/volume ratio) and kept under stirring for 24 h at room temperature. Then, methanolic extracts were filtered, the solid material was extracted again in the same way, and the filtered methanol from two different extractions was mixed and removed by vacuum distiller in a rotary evaporator to render methanol crude extracts.

### 4.2. Solvent Impregnated Resin (SIR) Methodology

Methanolic crude extracts were reconstituted with MeOH (50mL) and added 60 g of Amberlite^®^ XAD 4 20–60 mesh (Merck, Rahway, NJ, USA). The mixture was maintained by stirring for one to 300 rpm at room temperature to allow for to impregnation of compounds into the resin, and then was loaded into a glass column (40 cm high × 3 cm diameter) [26].

#### 4.2.1. Elution Protocol 1

The independent samples impregnated into resin were loaded into a column and eluted with 200 mL methanol (HPLC grade), obtaining fraction M1, and later, with 200 mL of deionized water (fraction W1). For all fractions, the methanol was removed and dried in a freeze-dryer under abovementioned the conditions. Three independent samples with three replicates each; E1, E2 and E3 were made for statistical analysis.

#### 4.2.2. Elution Protocol 2

Independent extract samples were eluted first with 200 mL of deionized water (fraction W2) followed by 600 mL of methanol (fraction M2), and the fractions were treated in the same form described above. Three independent samples with three replicates each; E1, E2, E3 were made for statistical analysis.

### 4.3. NMR Data

NMR spectra were acquired in a 600 MHz Agilent One NMR probe at the regulated temperature of 25 °C with PRESAT pulse sequence for water suppression. All samples were solved in DMSO-d_6_ and acquired with 128 transients, 30 s of relaxation delay, and an acquisition time of 3.4 seg for a digital resolution of 0.1. DDS was used as an internal reference for all spectra and for quantification purposes. For metabolite identification, ^1^H signals were compared with database of Chenomx software.

#### 4.3.1. Multivariate Analysis by NMR Data Processing

NMR spectra were processed with Mestrenova 14.2 version, an exponential apodization along T1 of 0.5 Hz was applied, phasing, baseline correction with a polynomial fit of order 3 along F1, and alignment and binning of 0.04 for all spectra. Multivariate analyses were performed using MetaboAnalyst 5.0 free software. Data matrix was normalized by median and Pareto Scaling.

#### 4.3.2. Principal Component Analysis (PCA)

Twenty samples from the Amberlite XAD-4 recovery flavonoids were randomly selected from methanolic and water fractions combined two protocols, ten samples from aqueous, and ten from methanolic fractions. PCA was applied to whole signals from the ^1^H NMR Spectra.

### 4.4. DIESI-MS Analysis

DIESI-MS were conducted on Bruker MicrOTOF-QII system by an electrospray ionization (ESI) interface (Bruker Daltonics, Billerica, MA, USA) operating in the positive and negative ion modes.

Each sample (1 mg) was resuspended in 1 mL of methanol, filtered through a 0.25 µm polytetrafluoroethylene (PTFE) filter, and diluted 1:100 with methanol to avoid saturation of the capillary and cone soiling. To improve ionization, 25 µL of pure formic acid were added to 475 µL of diluted sample [final concentration 5% (*v*/*v*) formic acid]. Diluted and acidified samples were directly infused into the ESI source and analyzed in negative mode. A constant volumetric flow rate (8 µL/min) was achieved using a 74900-00-05 Cole Palmer syringe pump (Billerica, MA, USA). Capillary voltage was set to 4500 V, and nitrogen was used as a drying and nebulizing gas, using a flow rate of 4 L/min (0.4 Bar) with a gas temperature of 180 °C. Continuous spectra were collected in a *m*/*z* range of 50–3000, with a total run duration of 1 min, a scan time of 10 s, and an interscan time of 0.1 s, producing six spectra per sample.

The mass spectrometer was operated at a resolution of 11,000 (FWHM) at mass 1622.0290 *m*/*z* in positive ion modes at a capillary voltage of 4500 V (positive) and 2700 V (negative). The spectrometer was calibrated with an ESI-TOF tuning mix calibrant (Sigma-Aldrich, Toluca, Estado de México, México).

Finally, precursor ion scans (MS/MS) were performed using negative and positive electrospray ionization (ESI^−^ and ESI^+^) with appropriate mass set. According to the obtained pattern, suitable fragments were analyzed by a Bruker Compass Data Analysis 4.0 (Bruker Daltonics), which provided a list of possible elemental formulas using Generate Molecular Formula Editor, as well as a sophisticated comparison of the theoretical with the measured isotope pattern (σ value) for increased confidence in the suggested molecular formula (Bruker Daltonics Technical Note 008, 2004). The accuracy threshold for confirmation of elemental compositions was established at 5 ppm.

### 4.5. Pharmacological Evaluation

Male Swiss Webster mice (25–35 g body weight) were obtained from the vivarium of the Instituto Nacional de Psiquiatría Ramón de la Fuente Muñiz. Animal care and use procedures were performed in compliance with the Mexican Official Norm (NOM-062-ZOO-1999), which is in concordance with the universal principles of laboratory animal care (NIH publication # 85–23, revised in 1985), and the local ethical committee approved the protocol (project number NC19127.0). Mice were housed in groups of 8 per cage on a 12 h reverse light/dark cycle (lights on; 12 h light/12 h dark) in a controlled temperature and humidity room. All experiments were done in the dark under a dim red light, from 10:00 to 14:00 h. Drugs were administered in a volume of 10.0 mL/kg body weight. Diazepam (DZ; 0.5, 1.0, 2.0, and 3.0 mg/kg), Imipramine (IMI; 25 mg/kg), and fluoxetine (FLX; 5 mg/kg/day) were used as control positive and administered by intraperitoneally (i.p.) route. Control groups were administered with vehicle (0.9% saline solution). Single doses of the experimental extracts were administered 30 min before starting the behavioral challenge.

#### 4.5.1. Anxiolytic-like Effect in the Elevated plus Maze Test (EPM)

This model is the gold standard behavioral model for detecting anxiolytic-like effects in rodents [21]. The apparatus consisted of two opposite open arms (30 × 8 cm), intersected (center platform) by two closed arms of the same dimensions, with 19-cm-high walls. The arms are connected to an 8 × 8 cm central square. The apparatus was elevated 55 cm above the floor.

At the start of the test, the mouse was placed in the center of the EPM, facing any of the open arms. the behaviors measured were the time spent in each arm (open arm; TOA, or close arm; TCA), these measures were expressed as a percentage respond calculated of follow form: % TOA = TOA/TCA + TOA × 100, and %TCA = TCA/TCA + TOA × 100. Additionally, Ethological behaviors, such as and time on the Central Platform (CPT) = [TOA + TBC] − 300; (300 s; test time, in such a way that TOA + TCA + TC = 300 s). the number of head dippings and stretches number were also measure. In this model, a decrease in the percentage of time spent on the open arms and an increase in the percentage of time spent on the closed arms with respect to the control group are both associated with low anxiety levels [27,28].

#### 4.5.2. Antidepressant-like Effects in the Tail Suspension (TST) and the Forced Swimming Test (FST)

The antidepressant-like effects were evaluated in two behavioral models, namely, the tail suspension (TST) and the forced swimming (FST) tests. In both, the passive behavior of immobility is induced. However, the TST is more sensitive to a wide range of antidepressants independently of their action mechanism.

Tail Suspension Test (TST)

In an independent experiment, mice were treated with a single administration of the experimental extract at 25, 50, 100, and 200 mg/kg or IMI (25 mg/kg), as described above.

They were held individually by the tail for 5 min and suspended 50 cm above the surface of a wooden box by adhesive tape placed 1 cm from the tip of the tail. Immobility behavior was measured for 5 min when the mouse remained passively hung and completely motionless, and this was reported as accumulated immobility time (s) [29].

Forced Swimming Test (FST)

Mice were individually placed into glass cylinders (height: 21 cm, diameter; 14.5 cm) containing 15 cm of water at 22 ± 1 °C and subjected to swimming for a 15 min period (pre-test session) 0 to stimulate the passive behavior of immobility. After 24 h they were subjected to a second swim session for 5 min (test session) [30].

To explore the antidepressant-like effects of Grapefruit extracts in the forced swimming test, two administration protocols were performed: (1) Independent groups of mice previously subjected to the pre-test session were administered a single dose of the naringin-rich extract (25, 50, 100, or 200 mg/kg) or imipramine (IMI) at 25 mg/kg, and after 30 min were subjected to the test session. (2) Independent mice groups were administered one daily dose of the naringin extract (25, 50, 100, or 200 mg/kg) or fluoxetine (FLX; 5 mg/kg/day) for seven days, on the sixth day, they were subjected to the pre-test swim session, on the seventh day, and 30 min after administration, they were subjected to the swim session test. Total immobility time (s) was measured in the last 3 min of the test session. All experimental sessions were videotaped and posteriorly measured by an observer unaware of treatment.

Open Field Test (OFT)

The locomotor activity of mice was measured in the open field to discard possible side undesirable or nonspecific effects of drugs. Open field device consists of an opaque-plexiglass box (40 × 30 × 20 cm) divided into 12 equal squares (11 × 11 cm). At the start of the test, the mouse was placed in any corner of the cage and videotaped over a 5-min period. The number of times the mouse crossed from one frame to another (counts number) and the number of times it stood on its hind legs (rearings number) were registered [31]. 

Hole-Board Test (HBT)

The hole-board apparatus is an acrylic box of 60 × 30 15 cm, with four equidistant holes (2 cm diameter) on the floor. Mice were placed at the center of the hole board, and the number of head-dips into the holes and the number of rearings (when the mouse stands up on its hind legs) were evaluated over a 5 min period. The animals were evaluated between 9:00 and 14:00 in a room lit by dim red light. Mice were administered with either DZ or the naringin-extract at 200, 4000, 800, and 1000 mg/kg 30 min before the test. In this paradigm, a decrease in head dipping number, the time of head dipping, and the number of rearings relative to the control group revealed a sedative action. At the same time, an increase in these variables is considered to have an anxiolytic-like effect [32].

Statistical analysis

Behavioral data that met the criteria of normality (Kolmogorov–Smirnov test) and variance equality were compared using a One-Way Analysis of Variance (ANOVA). Holm-Sidak’s test for multiple comparisons vs. control group was applied when the ANOVA showed a significant difference; *p*-values ≤ 0.05 were considered statistically significant. When data did not meet normality or variance equality criteria, a non-parametric analysis Kruskal Wallis analysis of variance on ranks (* *p* ≤ 0.05; ** *p* ≤ 0.01; and *** *p* ≤ 0.001) was used, followed by Mann-Whitney multiple comparison tests (* *p* < 0.05, ** *p* < 0.01, and *** *p* < 0.001. The SigmaPlot ver. 12.5 and Prism GraphPad statistical software programs were used to make the graphics and conduct the analysis. Data were represented as median ± standard error of groups of 8 mice each.

## 5. Conclusions

In this study, we perform an efficient, easy, and inexpensive method to naringin- and flavonoid-rich extracts from Grapefruit (*Citrus paradisi*) waste obtention that can be a valuable source of active principles with therapeutic potential for treatment of mood disorders.

The naringin-rich extract caused a clear anxiolytic-like effect in mice and reaped doses produced a robust antidepressant-like effect. Our results demonstrate the importance of bioprospecting studies of waste and agro residues with beneficial uses to human beings. Agro-residues may have a significant value, as nutraceuticals and drugs to treat anxiety and depression disorders.

## Figures and Tables

**Figure 1 molecules-27-08507-f001:**
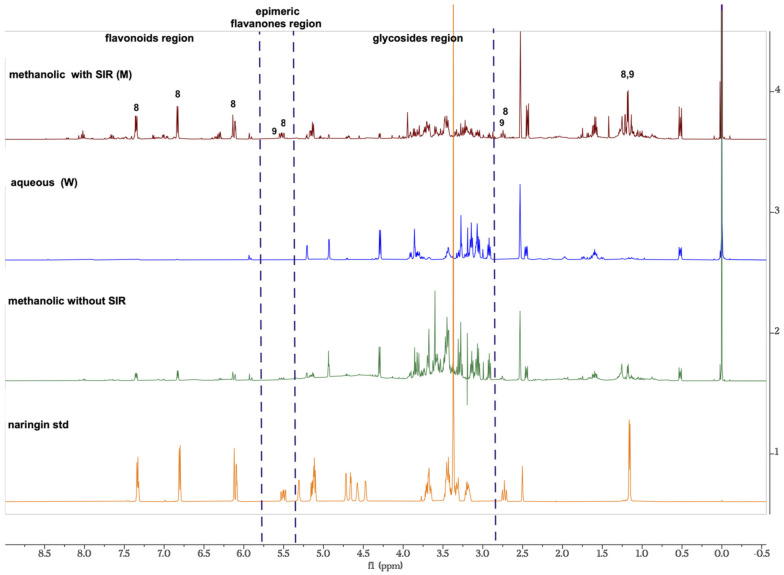
^1^H NMR proton spectra for comparison between naringin and methanolic extract without SRI treatment from Grapefruit and methanolic (M) and aqueous (W) extracts recovered from Amberlite XAD4 resin by protocol 1.

**Figure 2 molecules-27-08507-f002:**
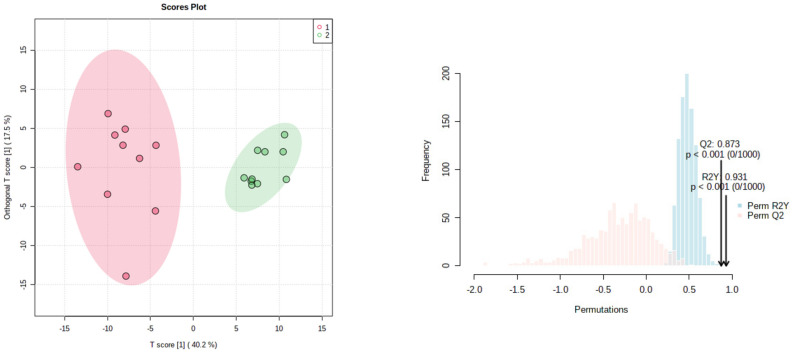
OPSL-DA, values of R^2^Y = 0.931 and Q^2^ = 0.873, underscore the reproducibility of the method. Pink cycles correspond to methanolic and green ones to water fractions.

**Figure 3 molecules-27-08507-f003:**
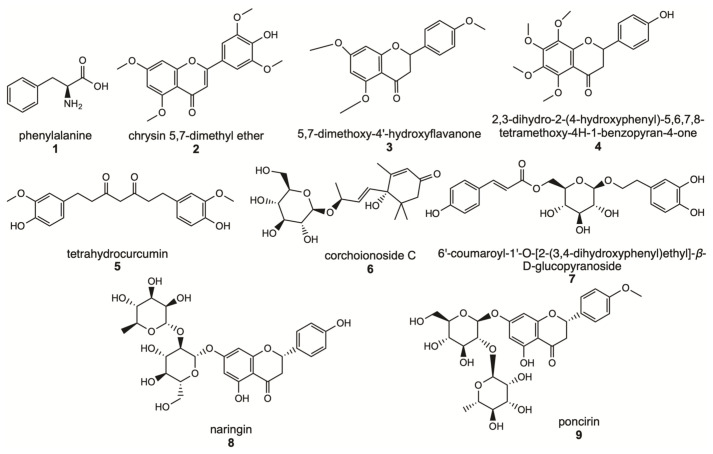
Main constituents of naringin-rich extract obtained by SIR method, protocol 1.

**Figure 4 molecules-27-08507-f004:**
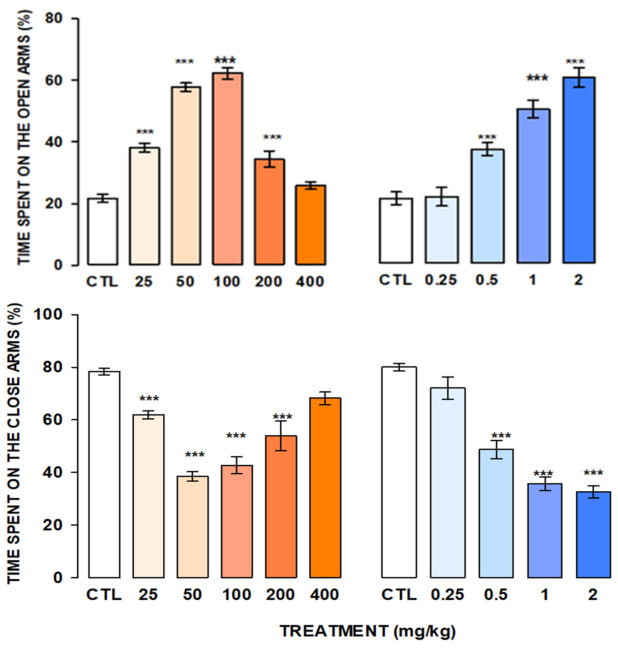
Anxiolytic-like effect of M1 at 25, 50, 100, 200, and 400 mg/kg (orange bars) and Diazepam at 0.25, 0.5, 1.0, and 2.0 mg/kg (blue bars) in the elevated plus maze test. Bars represent the mean ± standard error of the mean in groups of eight mice each. Data were analyzed by One Way ANOVA on Rank of Kruskal Wallis, followed by Mann-Whitney multiple comparison post-test vs. control group (CTL; white bars), *** *p* = 0.001.

**Figure 5 molecules-27-08507-f005:**
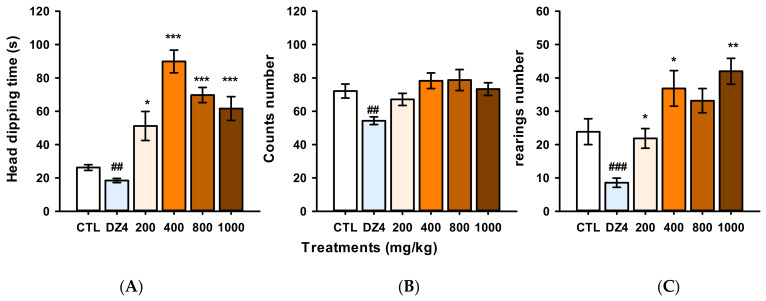
Sedative and anxiolytic-like effect of the naringin-rich extract and diazepam (DZ) in the Hole board test. (**A**) head dipping time (seconds), (**B**) Counts number, and (**C**) rearings number. Bars represent the mean ± standard error of the mean, of independent groups of eight animals each. Data were analyzed by One Way ANOVA on Rank of Kruskal Wallis, followed by Mann-Whitney multiple post-test in comparison with control group (CTL), * *p* ≤ 0.05, ** *p* ≤ 0.01, *** *p* ≤ 0.001 (increase), ## *p* ≤ 0.01, ### *p* ≤ 0.001 (decrease).

**Figure 6 molecules-27-08507-f006:**
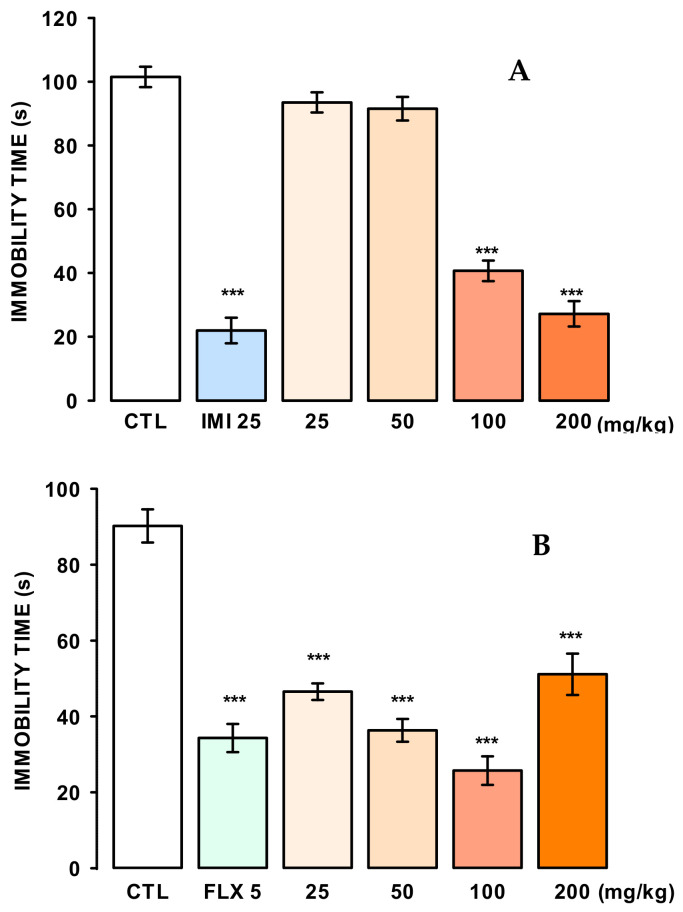
Panel (**A**) antidepressant-like effect of the naringin-rich extract (M1) and imipramine (IMI) in the Tails Suspension Tail Test. Panel (**B**) Antidepressant-like effect of naringin-rich extract and fluoxetine (FLX; 5 mg/kg). Bars represent the mean ± standard error of the mean, of independent groups of eight animals each. Data were analyzed by One Way ANOVA, followed by Holm-Sidak’s post-test concerning the control group (CTL), *** *p* ≤ 0.001.

**Table 1 molecules-27-08507-t001:** Fractions recovered from resin XAD-4 impregnation method in Protocol 1.

Fraction	E1 (grams) (M ± SD)	Recovery %	E2 (grams) (M ± SD)	Recovery %	E3 (grams) (M ± SD)	Recovery %
M1	11.43 ± 0.8	38.08	9.98 ± 0.35	33.3	10.40 ± 0.8	34.7
W1	0.75 ± 0.09	2	0.77 ± 0.24	2.6	0.77 ± 0.21	3

Data represent the mean (M) ± standard deviation (SD) of three extractions (E) with three technical replicates and % recovery from 30 g of dried grapefruit waste. The first elution was done with methanol (M1), second elution with water (W1).

**Table 2 molecules-27-08507-t002:** Fractions recovered from resin XAD-4 impregnation method in Protocol 2.

Fraction	E1 (grams) (M ± SD)	Recovery%	E2 (grams) (M ± SD)	Recovery%	E3 (grams) (M ± SD)	Recovery %
W2	11.41 ± 0.82	38	9.43 ± 1.55	31.4	10.36 ± 1.29	35
M2	1.3 ± 0.06	4.5	1.23 ± 0.46	4.1	1.49 ± 0.57	5

Data represent the mean (M) ± standard deviation (SD) of three extractions (E) with three technical replicates and % recovery from 30 g of dried grapefruit waste. The first elution was done with water (W2), second elution with methanol (M2).

**Table 3 molecules-27-08507-t003:** Main constituents of the methanol extract by DIESI-MS analysis.

	Name	[M + H] ^+^ _obs_	[M + H] ^+^ _exact_	Formula	Error (ppm)	mSigma	%RA EM	%RA M	%RA W
1	phenylalanine	166.830	166.0862	C_9_H_11_NO_2_	6.5	19.9	0.2	2.7	2.1
2	chrysin 5,7-dimethyl ether	283.0945	283.0964	C_17_H_14_O_4_	7.1	13.1	0.4	3.0	8.4
3	5,7-dimethoxy-4′-hydroxyflavanone	301.1099	301.1070	C_17_H_16_O_5_	−9.5	9.7	0.1	5.1	0.3
4	2,3-dihydro-2-(4-hydroxyphenyl)-5,6,7,8-tetramethoxy-4H-1-benzopyran-4-one	383.1157	383.1101	C_19_H_20_O_7_	−14.5	35.9	0.6	7.8	0.3
5	tetrahydrocurcumin	395.1425	395.1465	C_21_H_24_O_6_	16.1	19.4	0.4	4.2	5.8
6	corchoionoside C	409.1828	409.1832	C_19_H_30_O_8_	−1.1	23.2	0.4	1.8	1.8
7	6′-coumaroyl-1′-O-[2-(3,4-dihydroxyphenyl)ethyl]-*β*-D-glucopyranoside	463.1588	463.1598	C_23_H_26_O_10_	2.6	24.8	0.2	4.8	5.7
8	naringin	603.1656	603.1684	C_27_H_32_O_14_	−4.7	8.9	0.8	10.8	6.3
9	(2*S*)-poncirin	617.1764	617.1840	C_28_H_34_O_14_	−12.5	17.6	0.3	3.0	2.4

[M + H] ^+^ _exact_: exact Molecular Weight, [M + H] ^+^ _obs_: observed Molecular Weight, % RA: % Relative Area. Error [ppm]: Absolute value of the deviation between measured mass and theoretical mass of the selected peak in [ppm].

## Data Availability

The data presented in this study is available on request from the corresponding author.

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
