# Peer review of "Recovery of Naringin-Rich Flavonoid Extracts from Agroresidues with Anxiolytic- and Antidepressant-like Effects in Mice"

_molecules, 2022, doi:10.3390/molecules27238507_

Round 1

Reviewer 1 Report

The overview of the study is excellent. only minor check is required by the authors,

1. English editing is required for the minor spelling check.

2. Recheck the statistics of all the results where-ever required.

Author Response

The overview of the study is excellent. only minor check is required by the authors,

  1. English editing is required for the minor spelling check.

English was reviewed and improved

  1. Recheck the statistics of all the results wherever required.

We reviewed and corrected this mistake; thank you very much for your observation

Thank you very much for your criticism and the time dedicated to our work.

Reviewer 2 Report

a revision of the Discussion chapter is necessary, in order to clearly understand the results of the experiments. For example, it is mentioned: "a single oral administration with 200 and 400 mg/kg of the naringin-rich extract produced a clear antidepressant-like effect", and in the following sentence: "In contrast, a single treatment was not enough to produce this effect"

the References can be improved, of the 24 cited articles 6 are by the authors of the reviewed article

some of the keywords should be replaced: grapefruit/citrus, organic waste/agroresidues. It would be recommended to replace them with naringin, flavonoids

Author Response

Comments and Suggestions for Authors

a revision of the Discussion chapter is necessary, in order to clearly understand the results of the experiments. For example, it is mentioned: "a single oral administration with 200 and 400 mg/kg of the naringin-rich extract produced a clear antidepressant-like effect", and in the following sentence: "In contrast, a single treatment was not enough to produce this effect"

These sections were reviewed, and mistakes were corrected, it is improved to for a better understanding

Thank you for your criticism, the writ has improved remarkably

the References can be improved, of the 24 cited articles 6 are by the authors of the reviewed article

We have added references in the sections that needed them.

some of the keywords should be replaced: grapefruit/citrus, organic waste/agroresidues. It would be recommended to replace them with naringin, flavonoids

This was changed, thank you

Thank you very much for your criticism and the time dedicated to our work.

Reviewer 3 Report

In this manuscript, the authors have extracted different flavonoids from grapefruit waste and reported anxiolytic and antidepressant effects in mice. Based on the presented manuscript, the comments are as follows:

·         Abstract needs to rewrite and mention the scientific name of grapefruit.

·         In the introduction section, authors should mention the novelty of the present work.

·         Line no 46: plethora of pharmacological activities….The activity must be mentioned instead of using word ‘plethora’.

·         The prevalence of neuropsychiatric diseases must be discussed to show the relevance of current research.

·         Manuscript must be checked for grammatical mistakes.

·         Severe plagiarism is detected in many sections e.g. line no 388 to 508 are plagiarized.

·         Project number NC19127.0 is also associated with other studies published by authors.

·         References to few procedures are not mentioned.

·         Results of study are not clearly discussed.

·         The application of current study must be discussed with supportive references.

Author Response

Comments and Suggestions for Authors

In this manuscript, the authors have extracted different flavonoids from grapefruit waste and reported anxiolytic and antidepressant effects in mice. Based on the presented manuscript, the comments are as follows:

Abstract needs to rewrite and mention the scientific name of grapefruit.

We have added the botanical name of grapefruit to the abstract and introduction sections

In the introduction section, authors should mention the novelty of the present work.

this was highlighted in the introduction section.

Line no 46: plethora of pharmacological activities.

The activity must be mentioned instead of using word ‘plethora’.

This was corrected

The prevalence of neuropsychiatric diseases must be discussed to show the relevance of current research.

This topic was aborded in the introduction section.

Manuscript must be checked for grammatical mistakes.

This manuscript was reviewed and corrected

Severe plagiarism is detected in many sections e.g. line no 388 to 508 are plagiarized.

For DIESI-MS analysis, conditions of the methods employed in this research were previously established and tested for flavonoid detections, and it is difficult to change them. In the same sense, behavioral models are the tool for routine work; we paraphrased these sections and added references, and the data presented here is not the product of plagiarism. Project number NC19127.0 is also associated with other studies published by authors.

This is correct; this investigation belongs to the same institutional project.

 References to few procedures are not mentioned.

We enhance the bibliography.

Results of study are not clearly discussed.

We improve this to best understanding. The application of current study must be discussed with supportive references.

Thank you very much for your criticism and the time dedicated to our work

Round 2

Reviewer 3 Report

The authors have addressed all the concerns in the revised paper.